# "A labor of love": Integrating mental health and HIV care: Lessons from a multicountry mental health learning network

Modhurima Moitra[1] , Gloria Gonese[2], Madhuri Mukherjee[3], Belinda White[4], Dorraine Young[5] and Pamela Y. Collins[6] 

[1]Department of Psychiatry and Behavioral Sciences, University of Washington, Seattle, WA, USA; [2]Zimbabwe Technical Assistance Training and Education Center for Health, Harare, Zimbabwe; [3]I-TECH India, PL, New Delhi, India; [4]International Training and Education Center for Health, Port of Spain, Trinidad and Tobago; [5]Caribbean Training and Education Center for Health, Kingston, Jamaica and [6]Department of Mental Health, Johns Hopkins Bloomberg School of Public Health, Baltimore, MD, USA

## Perspective

**Keywords:**
learning networks; knowledge exchange; integrated mental health care; HIV

**Corresponding author:**
Pamela Y. Collins;
Email: pamela.collins@jhu.edu

## Abstract

Mental health conditions among people living with HIV (PLWH) are important to address as they adversely affect quality of life, impede adherence to HIV treatment and increase mortality. Planning for integrating mental health care in resource-limited HIV care settings requires substantial effort. Learning networks are a useful way to exchange knowledge between countries about best and current practices in planning mental health care for PLWH. This paper describes the launch of a mental health learning network within a global health implementing center and the lessons learned across participating members from six countries: the United States, Jamaica, Trinidad and Tobago, Zimbabwe, Malawi and India. Lessons learned from the learning network sessions spanned four broad domains: (i) the need for routine and integrated mental health data collection, (ii) developing standardized protocols to implement mental health care, (iii) adequate training and supervision of health care staff and (iv) prioritization of mental health care integration by program funders. We find that time and resource constraints can be barriers to shared leadership and sustainability of learning networks. Prioritizing learning networks as an important component of integrated HIV and mental health care programs is one of the potential strategies to ensure long-term continuity.

## Impact statement

Learning networks provide a useful platform for public health practitioners and researchers to exchange knowledge on current practices, implementation outcomes and challenges. Knowledge exchanged via learning networks may be more likely to be adopted into practice thus bridging the know-do gap. Though several learning networks exist in many fields of public health, we were unaware of international networks related to integrated mental health care in HIV care settings. This is particularly relevant given the growing number of national and global initiatives to address mental health conditions among people living with HIV (PLWH). Furthermore, as the evidence on best practices for integrated mental health care for PLWH continues to evolve, learning networks provide a unique and important platform for HIV care specialists and program implementing partners to exchange first-hand knowledge and experiences of identifying priorities and developing plans for mental health care. In this article, we describe the launch of the first multicountry mental health learning network within the International Training and Education Center for Health (I-TECH) at the University of Washington. We describe the network and its goals and summarize lessons learned from the network meetings in developing and implementing mental health care plans in HIV care settings. We also discuss potential strategies for long-term continuity and sustainability of learning networks.

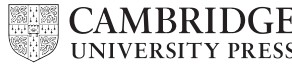

## Background

Addressing the mental health needs of people living with HIV (PLWH) is a global public health priority. Mental health conditions among people at risk for HIV and PLWH can reduce quality of life, increase risk of HIV infection, delay progress toward achieving viral load suppression, and subsequently, increase mortality (UNAIDS, 2022). Over the last 5 years, WHO and UNAIDS have emphasized the need for mental health and HIV integration through the launch of updated targets and the development of knowledge tools to encourage integration (UNAIDS, 2020; UNAIDS and WHO, 2022; WHO, 2022). The UNAIDS 2025 targets call for 90% of PLWH

and people at risk to be linked to contextualized services, including for mental health and social needs (UNAIDS, 2020). Donors have also recognized the importance of supporting the mental health of people affected by and living with HIV for the delivery of person-centered care that achieves programmatic goals. The United States President's Emergency Plan for AIDS Relief (PEPFAR) as well as the Global Fund for AIDS, TB and Malaria include mental health care as a key component of their upcoming multiyear strategy for HIV prevention and care (Collins and Kline, 2022; The Global Fund, 2023; US PEPFAR, 2023a,b). Planning for and initiating integrated mental health care approaches in already resource-constrained HIV care settings requires substantial effort as well as ongoing learning about what does and does not work in practice. In addition to appraising the evolving body of scientific literature, it is also important to engage in the exchange of knowledge between countries to learn of best practices and implementation challenges. Learning networks are a useful way to facilitate this exchange of knowledge and implement ideas shared by others experiencing similar issues.

Learning networks involve a process of convening individuals or organizations to engage in peer-to-peer learning for an area of common interest (Webster et al., 2019). They can accelerate program implementation and improvements in existing care approaches. In particular, program specialists and clinicians are more likely to adopt and implement ideas learned from such networks than from other didactic training methods (Webster et al., 2019). Learning networks can assist in the exchange of knowledge and ideas for important global public health issues such as preventing mother-to-child transmission of HIV, improving adolescent HIV prevention and care, or enhancing nutritional outcomes for newborns (Barker et al., 2019; Webster et al., 2019; Sturke et al., 2020). Collaborative approaches to learning improve standards of care and have been applied explicitly for quality improvement to achieve better outcomes in mental health, maternal and child health, malaria, tuberculosis and other conditions in a diversity of health care settings globally (Franco and Marquez, 2011; Nadeem et al., 2014). However, to our knowledge, no published reports of cross-country efforts for learning describe support for the integration of mental health and HIV care. In 2022, the International Training and Education Center for Health (I-TECH) at the University of Washington piloted a multicountry mental health learning network to facilitate knowledge exchange and capacity building among I-TECH network sites on how to integrate mental health care in HIV care settings. The learning network launched by two of the authors (PYC and MM) followed a 2021 cross-site assessment to determine how mental health needs of people at risk for and living with HIV might be addressed. This paper reports lessons learned and shared across participating countries through the learning network. We also report participant perspectives on the feasibility, acceptability and utility of a learning network.

## Overview of I-TECH

The International Training and Education Center for Health (I-TECH) is a global health implementing center at the University of Washington with activities in more than 20 countries. I-TECH collaborates with local partners to build and sustain strong national health systems, strengthen the public health workforce and utilize data-driven approaches to implement public health interventions in Africa, the Caribbean, Central Asia, Eastern Europe, Latin America and South Asia (University of Washington, 2023). The Center primarily works across six focus areas: health systems strengthening; health workforce development; health leadership and management; prevention, care and treatment of diseases; global health security, one health and public health surveillance; and implementation science and evaluation (University of Washington, 2023). I-TECH designs and implements projects to adhere to its 'Partnership Model', which encourages the transition of technical services and assistance to country ownership, allowing for long-term and sustainable continuation and scale-up of programs (University of Washington, 2023).

I-TECH comprises a network of country offices linked to the University of Washington as well as independent "partner organizations" that evolved from former I-TECH country offices. It is a PEPFAR implementing partner. In this capacity, I-TECH country offices and partner organizations work closely with ministries of health, national AIDS coordinating bodies and civil society organizations to carry out activities aimed to control the AIDS epidemic (University of Washington, 2023). In some countries, I-TECH activities center on supporting national training activities, designing and implementing health information systems, or guideline development and implementation. In others, I-TECH and partner organizations support site-level HIV service delivery, clinical mentoring and other aspects of HIV prevention, care and treatment. Where I-TECH teams deliver facility-based services, teams work in designated regions of a country and in facilities specified by the project funder and national health ministries. In the context of HIV care, I-TECH team members have reported a growing need for mental health and substance use interventions in HIV services.

## I-TECH mental health learning network

We planned a 6-month pilot of a mental health learning network that extended to eight sessions delivered over 10 months (September 2022 – June 2023) due to members' interest. The I-TECH center director, working with a UW post-doctoral fellow, invited participation from sites across the I-TECH network and from interested I-TECH faculty and staff at the University of Washington. The invitation to the learning network was distributed through the director's routine email updates in the weeks prior to the launch. Participants who volunteered to join the network included I-TECH country team and partner organization members who constituted site leadership, psychologists, counselors, technical team leads and faculty. The network included 16 participants representing six countries: the United States, Jamaica, Trinidad and Tobago, Zimbabwe, Malawi and India.

PEPFAR's annual country and regional operational plan (COP/ROP) guidance for PEPFAR-supported countries encourages implementers to address behavioral health needs as part of person-centered HIV care and to integrate national plans and programs. The Fiscal Year 2024 Technical Considerations recommend that "teams should prioritize behavioral health interventions when they demonstrate a substantial impact on overall program success, and support interventions that are evidence-based" (US PEPFAR, 2023a). Notably, PEPFAR guidelines also caution that the program funding is not able to fully support specialist mental health service delivery, and collaboration and coordination with existing mental health services is also recommended (US PEPFAR, 2023a,b). Each of the learning network country teams had PEPFAR-funded work plans that included mental health activities. These countries also had a dearth of specialist mental health providers for the population (Table 1).

**Table 1.** Selected mental health and HIV data for countries represented in the learning network

| | Mental health service data | | | HIV cascade data (UNAIDS 2023) | | |
|---|---|---|---|---|---|---|
| Country | Psychiatrists (per 100, 000 pop) | Psychologists (per 100,000 population) | Community-based mental health facilities | % of PLWH who know their HIV status | % of PLWH who know their status and access HIV care | % of PLWH who know their status, are accessing treatment and are virally suppressed |
| India | 0.75 | – | – | 79 | 86 | 93 |
| Jamaica | 0.75 | 0.54 | 137 facilities | 92 | 54 | 91 |
| Malawi | 0.01 | – | – | 94 | >98 | 94 |
| Trinidad and Tobago | 2.22 | 1 | 29 facilities | – | – | 93 |
| Zimbabwe | 0.01 | 0.02 | 1 hospital-attached facility | 95 | >98 | 95 |

*Sources*: Garg et al., 2019; Scotland-Malawi Mental Health Education Project et al., 2020; WHO 2022a,b,c,d; UNAIDS, 2023.

Network members convened virtually for monthly meetings lasting 90 min. The meetings were coordinated by the post-doctoral fellow and co-led by one or more participants who delivered Power-Point presentations. At the end of each meeting, members typically volunteered to prepare a presentation about their site activities for the subsequent meeting. The monthly network meetings included presentations of ongoing work to expand, enhance or initiate mental health and psychosocial support interventions in I-TECH network field sites. Notably, teams were at different stages of mental health integration activities. Thus, the monthly presentations described proposed areas of work, as well as areas of progress and challenges that were discussed with participants. In Table 2, we summarize the topic areas, along with strengths and challenges

discussed. Toward the conclusion of our pilot period, we conducted an informal focus group to discuss feasibility, acceptability and utility of the learning network. Table 3 displays a summary of feedback provided by attending members.

## Mental health learning network goals

Five primary goals were identified at the first network meeting. All goals were reviewed and approved by attendees.

1. *Foster cross-site learning*: Support learning to develop and/or implement international and national guidelines, provide technical assistance to ministries of health, deliver clinical mentoring

**Table 2.** Areas of learning presented by network members

| Topic | Description | Strengths | Challenges |
|---|---|---|---|
| Zim-TTECH/Zimbabwe Partnership to Accelerate AIDS Control (ZimPAAC) mental health initiative | To capacitate lay cadres to screen for common mental health problems among PLWH<br>To provide first-line response using the Listen, inquire, validate and enhance safety (LIVES)/problem solving therapy (PST) approach | Develop training and support plans<br>Introduce new registers and leverage existing HIV registers for improved data collection<br>Develop standardized flowchart and protocol for screening, triage, and referral for further treatment | • No dedicated mental health funding<br>• Staff attrition<br>• Weak referral systems<br>• Competing priorities within HIV program<br>• Mental health outcomes not aligned with funder outcomes |
| I-TECH psychosocial evaluation and MHPSS protocol development in Trinidad and Tobago | To assess mental and substance use disorders within HIV care sites<br>To strengthen delivery and oversight of psychosocial services for PLWH | Identified mental disorders prevalent within HIV care sites<br>Identified behavioral and socioeconomic risk factors presented among clients<br>Drafted protocol for integration of mental health and HIV care | • High patient caseloads<br>• Lack of coordinated referral systems<br>• No system-wide service integration of HIV and mental health care<br>• Limited training for staff |
| I-TECH India planned pilot intervention on mental health integration | To promote an integrated model for screening, referral, and counseling of mental health-related disorders among PLHV | Develop the integration plan with district mental health program (DMHP) for screening and referral at site and district level. Explore linkages with the DMHP facilities and virtual platforms like Tele MANAS/E-Sanjivani<br>Conduct workshops on mental health among PLWH<br>Develop training, support, and data collection plans with district officials | • Linkage to care<br>• Retention in care<br>• Achieving optimal symptom and care management<br>• Lack of awareness, stigma, segregation |
| Mental health and psychosocial support in HIV care at Lighthouse Trust ART clinic in Malawi | To provide an overview of ongoing mental health support at HIV care clinics | Multilevel mental health interventions including counseling, medication, and referrals.<br>Collaborative partnership with Ministry of Health along with ministry-level investments | • Limited mental health staff<br>• Need for updated monitoring and evaluation tools<br>• Ambivalence about psychosocial services by some healthcare workers |

**Table 3.** Feedback on feasibility, acceptability and utility of learning networks

| Theme | Summary of feedback received |
|---|---|
| Feasibility | Participating network members agreed on the feasibility of attending and/or co-leading any future sessions to share ongoing or recently completed work. The importance of members from different country sites leading sessions was emphasized as it serves the goals of cross-site collaboration and exchange of knowledge |
| Acceptability | Participating members unanimously agreed on the acceptability of the content and format of these network meetings. Various country teams were at different phases of developing and implementing mental health care protocols. Therefore, these meetings provided a unique opportunity for members to learn from one another in real-time as they made progress toward improving mental health care at their respective sites |
| Utility | Participating members agreed on the utility of the meetings particularly because of the similarity of challenges faced across sites and the opportunity to receive and make useful recommendations. Presentations by network members provided useful lessons and takeaways that could potentially be adapted to local settings. Suggestions were made to develop a resource hub that has now been launched and designed for integration into the I-TECH intranet |

and training, select sustainable interventions for implementation and monitor and evaluate programs.

2. *Review of the current literature*: Review the scientific and gray literature in order to stay informed of the latest evidence on best practices and global actions related to integrated mental health and HIV care that could potentially inform our work.
3. *Generate research questions:* Identify problem areas for translation into practice-based research questions, seek research funding and expand the field's knowledge.
4. *Strengthen dissemination of our work to diverse audiences:* Discuss methods of sharing our work through different media and tailor information to be accessible to a wide range of audiences.
5. *Global Mental Health Partnerships:* Build and foster communities of partnerships and collaborations. Liaise with other internal and external centers and with members of the University of Washington Consortium for Global Mental Health.

## Lessons learned and challenges identified

Written notes were taken for all learning network Zoom meetings and 5 of 8 sessions were audio recorded. Audio/video recording began with session 2, and two sessions were not recorded due to technical issues. Although the pilot did not include formal data collection, we reviewed session notes and video files, and we dedicated one meeting to discussing the benefits and challenges associated with the learning network. We organize lessons learned from the learning network by four broad themes that were most commonly highlighted during these meetings: (1) Routine Mental Health Data Collection, (2) Development of Standardized Protocols for Mental Health Care Integration, (3) Training and Supervision of Healthcare Staff and (4) Prioritization of Mental Health Care Integration. Next, we describe how participants exchanged information within and outside of the network sessions; we assess the achievement of planned goals, successes, and challenges of conducting the learning network; and we discuss future directions for learning networks.

### Routine mental health data collection

Network members emphasized the need to implement clinical data collection mechanisms that tracked clinically relevant mental health care outcomes. Across sites, data from screening tools such as the Patient Health Questionnaire–2 (PHQ-2) items or PHQ-9 items, Generalized Anxiety Disorder-2 (GAD-2), the Shona Symptom Questionnaire-14 (SSQ-14 in Zimbabwe) were the most commonly collected type of mental health data. In an early team presentation, one network member noted that identification of mental health conditions occurred haphazardly, though more identification occurred in facilities where providers had received training in the World Health Organization's Mental Health Gap Action Program (mhGAP). In Trinidad and Tobago, proposed screening instruments also included the CAGE questionnaire for alcohol use disorder. In several countries, screening for mental health conditions (especially depression) was encouraged through national HIV care guidelines. In Jamaica, where a recent study showed that 65.6% of PLWH screened positive for at least one psychiatric disorder (Beckford Jarrett et al., 2018), psychological interventions and psychiatric care were offered in some HIV service settings. Mental health data were collected via brief screening tools (e.g., PHQ-2 and the GAD-2) as well as more detailed clinical assessments. Psychologists in the National HIV/STI program were required to submit monthly reports on the number of clients seen, their demographics and the nature of the mental health conditions. The information was collated by the Strategic Information (SI) unit in the HIV/STI/TB Unit of the Ministry of Health and Wellness but was not published in the SI unit's quarterly bulletin. The Mental Health Initiative in Zimbabwe proposed to introduce new data collection registers as well as leverage the use of existing registers to better align data collection with proposed clinical interventions. A pilot intervention planned in India proposed to develop data monitoring tools that will help track mental health indicators and evaluate outcomes of the program.

In group discussion across two sessions, participants noted that in most sites, neither mental health screening nor clinical outcome data was routinely integrated into the electronic HIV health information systems. In some sites, paper registers at clinical facilities held mental health assessment data and tracked longer term clinical status of people receiving mental health interventions. While essential for clinical care, in the absence of integrated electronic systems, the introduction of new paper registers to collect additional data was likely to create administrative challenges, including additional workload and time spent on documentation by clinical staff. The utilization of registry data typically depends on the availability of staff time to verify data accuracy for clinical decision-making and conduct analyses for broader facility and health system program decision-making. The shared challenges of routine mental health data collection included the lack of integrated mental health registers, gaps in documenting client mental health outcomes and challenges in transitioning from paper-based data collection to electronic modes of data collection or consistent utilization of them where they existed.

### Development of standardized protocols for mental health care integration

The need to develop standardized protocols in order to maintain a consistent approach to screening, diagnosis and care for mental health conditions was shared by several network members. To accomplish this, several teams developed plans to implement protocols that were shared at the network meetings. In its protocol

for expansion of services, the Zimbabwe Mental Health Initiative planned to utilize brief screening tools (e.g., PHQ-2, GAD-2 and the Shona Symptom Questionnaire-14 (SSQ-14)) to triage clients for detailed assessments, following which they would be referred for further care if needed.

The HIV care sites in Trinidad and Tobago referred clients in need of mental health care to services in the mental health system (outside of HIV care). PLHIV could access outpatient care from Mental Health and Wellness Centers or psychiatric inpatient admission or rehabilitation facility care depending on the severity of their conditions. The patients would be required to undergo a separate enrollment process to access mental health services. HIV treatment and care site staff noticed that patients referred for this type of care often failed to follow through with the enrollment process due to several factors, including fear of stigma and discrimination related to their HIV status. To address this concern the Ministry of Health's HIV/AIDS coordination unit intended to pilot a Mental Health and Psychosocial Support protocol, which would provide the integrative services of identifying (via routine screening) and treating mental illnesses among PLHIV at the HIV treatment and care sites. The development of this protocol occurred during the course of the learning network pilot.

In Malawi, national HIV guidelines launched in 2022 included specifications for managing mental health conditions in HIV care. Network participants estimated that providers who received training on the guidelines were equipped to provide screening and referral commensurate with their roles in the system, though a majority of sites still needed training at the time of the learning network discussions. Ongoing mental health care in Malawi included care from multiple teams, including a psychosocial team, retention team and a clinical team that worked together to assess for mental disorders, provide care and ensure retention in care. In Jamaica, peer navigators provided additional support during the screening and triage process, following which clients were referred to further care as needed.

Common challenges discussed by network members included poor coordination and lack of appropriate referral resources, lack of tracking systems for referrals between HIV and mental health services and lack of access to clinical information gained by a referral. In one setting, participants described psychosocial services being perceived by other healthcare workers as having low value. Therefore, this may pose challenges to developing and adhering to standardized protocols for mental health care.

### Training and supervision of healthcare staff

Network members emphasized the need for appropriate training and supervision of healthcare staff to deliver mental health interventions. Four of the five country teams entered the learning network with experience providing mental health support or training and mentoring providers to deliver mental care in HIV care. Three of the participating teams were planning to roll out mental health care training modules for healthcare staff as well as designate task-shifted roles to specifically address different components of mental health care. The Mental Health initiative in Zimbabwe aims to expand training of lay cadres and healthcare workers to screen and manage common mental health conditions. It also aims to provide regular mentorship and support along with weekly observations and feedback sessions to support newly trained staff. The team in Trinidad and Tobago described their psychosocial program assessment comprising semistructured interviews and focus groups aimed to inform the strengthening of mental health service delivery.

A series of mental health focused trainings were being conducted at the Jamaican sites. In group discussions, team members unanimously acknowledged the challenges of limited staff, high caseloads and limited resources for training and ongoing supervision that are needed to provide adequate and high-quality mental health services to PLWH.

### Prioritization of mental health care integration

The successful integration of mental health services in HIV care settings requires funding and policy-making bodies to explicitly prioritize these efforts in the form of laying out guidelines and making additional investments (Kaaya et al., 2013). Team members described the progress made in several ongoing programs. The Ministry of Health, World Health Organization (WHO) and the Friendship Bench partnered with Zimbabwe Technical Assistance and Training Education Centre for Health (Zim-TTECH) to expand integrated mental health service delivery in HIV settings. These activities occurred in the context of larger mental health service expansion in Zimbabwe, which is one of the nine countries participating in the WHO Special Initiative for Mental Health, which aims to scale up mental health services in priority countries (Kemp et al., 2022). The Malawi Ministry of Health has actively invested in scaling up mental health services and developed guidelines for integrating mental health into HIV care. Jamaica's MOHW planned to scale up mental health integration by training all primary care physicians in mhGAP and strengthening mentoring and consulting services in all health regions in the next programming year. Mental health is a core component in India's technical guidelines for care and treatment of PLHIV, and integration activities were occurring gradually as was attention to care for other noncommunicable diseases in HIV care. The focus here is to link services and referral mechanisms for mental health needs of PLHIVs with district mental health facilities, under the mainstream health program.

In group discussions, participants lauded these efforts. However, as a result of competing priorities within existing HIV care programs, there is less effort to report on mental health indicators. Funding for scaled up mental health delivery also continues to be limited in most countries.

### Informal exchange of knowledge

Learning network participants unanimously agreed that the sessions were informative and provided insights into how they might approach the contextual specificities of HIV and mental health service integration. Learning occurred through probing and questioning methods that participants used, leading to shared problem solving as well as consultation within and between sessions. We share example quotations that illustrate these exchanges.

"One positive…(of) what you are proposing is the collaboration with various partners, which is key and critical. I was curious to know if there was central funding going to these various partners… (and) whether the priorities of these…partners is aligning with the model you are proposing."

Some participants delivered multiple presentations over the course of the pilot project, demonstrating progress in their plans for integration and showing what they learned from others.

"I have learned so much about what is being done, especially in (country) and what has been really effective with regard to providing mental health services, not specifically by mental health professionals, but by lay persons who have been trained and their

capacity has been strengthened…It's something I've spoken to the Ministry of Health here about adapting and strengthening the capacity of our (cadre of lay providers)."

Others sought consultations outside of the learning network sessions and alerted members of their intentions to do so. "I'm looking forward to learning and sharing. Many of the countries have already demonstrated success in mental health integration… maybe look at opportunities where we can reach out to you individually to learn more."

Participants acknowledged the difficulty of the work due to the uncertainty of sustained funding, national shortages of health care providers and underresourced mental health systems. One network member commented, "…it's really a labor of love."

### Goals and process learning

We launched the learning network aiming to foster cross-site learning, review of the current literature, generate research questions, strengthen dissemination of our work to diverse audiences and enhance global mental health and other partnerships. We successfully fostered cross-site learning, as perceived by the participants. We published a review paper that reviewed global mental health literature on which several members of the learning network were coauthors; we discussed current research findings relevant to integrated care in selected learning network sessions. Team members from the United States, Jamaica, Trinidad and Tobago and Zimbabwe participated in a World AIDS Day webinar focused on mental health integration hosted by the Mental Health Innovation Network, a global mental health initiative, enabling the team to meet its goal of dissemination. The group did not generate formal research questions or broaden collaborations during the pilot phase.

We identified several lessons associated with running and conducting a multicountry learning network. First, designating a fixed six-month term for the pilot of the network proved to be a useful way to garner interest from members to participate and this eventually extended to 9 months based on members' interest. Second, the learning network provided a useful source of peer support for professionals working on mental health care, an area that is underresourced in the members' countries, thus creating challenges that extend to HIV care. Third, a learning network coordinator with dedicated time and effort was assigned for this role, whereas the majority of learning network participants integrated these meetings into their afterwork or workday activities. Consequently, though the learning network was conceptualized as a horizontal structure with the aim of participants eventually taking ownership of the network sessions, organizational responsibilities (e.g., setting meeting agendas, identifying presenters) largely remained with the coordinator. Not surprisingly, we noted that time and resource constraints can be a barrier to shared leadership of a learning network. This is also linked to the challenge of sustaining such networks as well as monitoring and assessing data on shared network goals in the long term.

### Future directions

The scale, overarching goal and resources of a learning network will shape how frameworks for operation are selected, implemented and sustained. Quality improvement frameworks are frequently employed through learning networks (Nadeem et al., 2014). The large, complex, quality improvement-centered multicountry learning network described by Webster et al. documented methods of

knowledge exchange, key areas of learning and associated outcomes in its evaluation (Webster et al., 2019). This team also determined level of impact and cost associated with the variety of knowledge exchange methods implemented (Webster et al., 2019). The authors emphasize the importance of country ownership and co-design, clarity of roles, a designated group for knowledge management, relationship-building, continuous evaluation and adaptation and the use of data and stories for learning for successful large-scale learning networks (Webster et al., 2019).

Many of these elements also ring true for smaller networks as they launch. We recommend using preparatory sessions to determine the overarching goal(s), level of evaluation (e.g. network processes, site-level or above-site practice change) and evaluation framework with agreed upon measures. Similarly, bolster networks by incorporating them as one of the funded core components of integrated HIV and mental health care programs; allot fixed durations of participation and opportunities for in-person exchange to increase member engagement; regularly collect data on types of exchange, areas of learning, how knowledge is utilized and the associated outcomes, acceptability and utility of the network; and encourage members to track shared goals and long-term continuity.

### Conclusion

Learning networks are a useful way to facilitate knowledge exchange in rapidly evolving fields of practice such as integrated mental health and HIV care. The I-TECH mental health learning network provided peer supervision for a diverse set of professionals engaged in or interested in HIV and mental health service integration, space for vetting new ideas and expert consultation. The pilot initiative achieved its goal of exposing practice-based professionals to research and dissemination activities. Future long-term initiatives like these can sustain this learning process and encourage rigorous evaluation and wider dissemination of information to stakeholders interested in integrating mental health care into HIV settings.

**Open peer review.** To view the open peer review materials for this article, please visit http://doi.org/10.1017/gmh.2024.6.

**Acknowledgements.** We would like to thank the I-TECH Mental Health Learning Network members for their participation: Vivian Bertman, Rajeeb Dey, Mohit Goyal, Joel Holding, Christine Kiruthu Kamamia, Lewis Masimba, Tameca Dempster-Mattocks, Martin Maulidi and Katherine Wilson. Dr. Collins was executive director of I-TECH and a professor at the University of Washington at the time of the learning network's pilot activities.

**Author contribution.** Conceptualization: M.M., P.Y.C.; Funding acquisition: P.Y.C.; Project administration: M.M., P.Y.C.; Supervision: P.Y.C.; Writing–original draft: M.M.; Writing–review and editing: G.G., M.M., B.W., D.Y., P.Y.C.

**Financial support.** This work was supported in part by the National Institute of Mental Health, P30 MH123248 and the Health Resources and Services Administration, U91HA06801.

**Competing interest.** The authors declare none.

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
