## [Reviewer Report]

Dear Dr. Chibanda and Dr. Bass,

My colleagues and I would like to submit our manuscript titled “Integrating mental health and HIV care: Lessons from a Multi-Country Mental Health Learning Network” to be considered for publication as a perspectives paper in the Global Mental Health journal. In this manuscript, we discuss the importance of learning networks as a unique way for public health professionals to exchange knowledge about best and current practices in improving health outcomes. This is particularly relevant for integrating mental health care in HIV care settings. Planning for integrating mental health care in resource-constrained HIV care settings requires substantial effort. Learning networks may provide a useful platform to bridge the know-do gap and facilitate exchange of ideas and knowledge between practitioners facing similar challenges. We describe the launch of the first multi-country learning network by the International Training and Education Center for Health (I-TECH) at the University of Washington. We also provide a summary of lessons learned from the members participating in our network sessions. 

We hope the editorial team will consider our manuscript for publication. We confirm that this manuscript is not published or under consideration for publication elsewhere. All authors approve with the current version of this manuscript. We look forward to receiving your feedback at the earliest. 

Sincerely,

Modhurima Moitra, PhD

---

## [Reviewer Report]

This study reports on lessons learned from a multi-country learning network focused on integrating mental health care into resource-limited HIV care settings. Given the importance of integrating mental health services into other clinical services to expand access to care and the elevated prevalence of mental health conditions among PLWH, the projects represented within this learning network each represent an important focus. Further, sharing lessons rather than re-inventing the wheel promises to help increase progress in global mental health and the intentions of this network are much appreciated. I have made some suggestions below to further strengthen the reporting on this network, primarily focused on clarifying the practices of the network itself.

- Line 45: Please briefly describe any national or sub-national learning networks on the integration of mental health and HIV care. Were any choices made differently for this network due to its goals or participant needs?

- Line 77: How were learning network participants selected?

- Line 80: Is there an explicit framework for obtaining feedback or input from other members that learning members use? Additionally, could you include more specific examples of how the interaction between participants positively impacted a given project? Currently the examples listed in the lessons section are all presented independently.

- Line 87: How will the learning network know if its goals have been met? How are goals monitored?

- Line 107: Was there a framework used for identifying the lessons from the network itself? It would be helpful to learn more about how these lessons were identified.

- Line 173: reference is missing.

---

## [Reviewer Report]

Thank you for the invitation to review this very well-written perspective piece on a topic of critical importance. The learning network initiative described in the manuscript represents a valuable approach to knowledge sharing among international sites and the authors identified important implications of the work. I believe this perspective can make a valuable contribution to the literature and provide minor edits below that I believe will improve the work prior to publication.

Line 38, “exchange of knowledge…”

The descriptions of themes 1 and 2 should highlight the need to prioritize clinical utility in mental health data collection. While I agree that improved data collection is needed to “track mental health indicators and evaluate outcomes”, the most urgent need for these data at the clinic level is to inform immediate assessment, care delivery, and referral. I would like to see some acknowledgement of the relationship between screening data (brief, easy to administer/low provider burden, clinically useful) and registry data (likely longer, require additional staff to collect and maintain, but still important that these are available to inform clinical care when warranted).

Theme 2: What kinds of brief screening tools are being considered? Are these brief symptom checklists and, if so, what symptoms are being emphasized? Or are sites considering more transdiagnostic assessments or global assessments of distress? Again, what aspects of mental health are/should be emphasized in screening?

What is the availability of psychotropic medication at the partnering sites and where does this fit into their priorities for improving mental health care?

What were some of the reasons for a perceived “lack of value assigned to psychosocial services by other healthcare workers”? Brief mention of these challenges will set the tone for the training priorities described in Theme 3.

Lines 188, remove the word “was”

---

## [Reviewer Report]

Multi-Country Mental Health Learning Network is a very useful and innovative approach to increase capacity of staff who provide health care for PLWH and identify gaps where interventions can effectivly improve quality of care for PLWH. In the manuscript there are no data (quantitative or qualitative), that provide evidence on the lessons learned. Also, lack of information that allows a reader to understand challneges on a systemic level of health care services in each country (number of clinics that provide services, number of doctors, number of patients, workload and types of personnel who can provide different level services, system of registrations and referral for mental health treatment).

---

## [Reviewer Report]

Background:Please provide a more detailed explanation about what a learning network is including the forms it might take. In addition to the citations, an example of the practical use of learning network and how it has been used either in low resource settings or for another infectious disease would help further put this into perspective.

Line 23, please fix the abbreviation of PWLH to PLWH

line 56, how long has I-TECH been operational in the partner countries, and a bit more detail on how it works eg who it works with in these countries , are these government health facilities if so at what level, additionally a bit more information on how its partners are chosen.

Line 117, is routine mental health data collection a mandate/ enforced by the governments in these countries. If so it would be good to state this upfront-provides better perspective for the reader.

line 119 could you provide more specific information about how many counties in your sample did not routinely intergrate their mental health data

line 120 if possible could you describe the type of data and how similar or different it was across the sites.

line 122 please provide more detail about the type of administrative challenges

line 134 the term mental health problems broadens the scope significantly and may sound stigmatising to some readers. perhaps terms such as conditions would be more appropriate? additionally try and use a consistent term through the manuscript.

line 152: again more specific, how many teams exactly are planning this training

line 170 :in the examples below is it possible to provide some detail on how far their progress has gone.

line 172 :please include the reference.

---

## [Reviewer Report]

We appreciate that the research community could benefit from reading about the lessons learned from a multi-country mental health learning network, however, no data has been included. We would like to request that some quantitative or qualitative data be included to support this manuscript.

---

## [Reviewer Report]

The authors have added a considerable amount of detail into this paper - on how the learning network operated and on specific outcomes of this learning network - and this is extremely helpful. I would still find it helpful if future directions and evaluation of the network, rather than just description if it, were emphasized more. What types of frameworks can best guide development of future learning collaboratives on this topic? How could members be included and engaged going forward to increase cross-site collaboration, or equity, or some other goal? What are the types of data one should collect in the future so evaluation can be more formal? Reflecting on these types of questions would help make this more generalizable to others and less descriptive of a specific project.

---

## [Reviewer Report]

My comments from the first review have been fully addressed. I believe this manuscript will make a valuable contribution to the literature.